# Resuscitation in Community Healthcare Facilities in Israel

**DOI:** 10.3390/ijerph18126612

**Published:** 2021-06-19

**Authors:** Irena Zherebovich, Avishay Goldberg, Amir Ben Tov, Dagan Schwartz

**Affiliations:** 1Department of Emergency Medicine, Faculty of Health Sciences, Ben-Gurion University of the Negev, Beer-Sheva 841051, Israel; zuerebov_i@mac.org.il; 2Maccabi Healthcare Services, Tel-Aviv 6812509, Israel; amir.bentov@gmail.com; 3Department of Health Policy and Management, Faculty of Health Sciences, Ben-Gurion University of the Negev, Beer-Sheva 841051, Israel; avishy@bgu.ac.il; 4Sackler Faculty of Medicine, Tel-Aviv University, Tel-Aviv 6997801, Israel

**Keywords:** resuscitation, out of hospital cardiac arrest, community healthcare workers, healthcare management

## Abstract

Background: Out-of-hospital cardiac-arrest (OHCA) is a major public health challenge. Community health care providers (CHP) may play an important role through early identification, basic life support and defibrillation. Few studies have evaluated the incidence and characteristics of OHCAs initially cared for by CHP, most finding improved survival. This study combined CHP treated OHCA case analysis, with assessment of provider resuscitation preparedness. Methods: An analysis of all CHP initiated resuscitations in a large Health Maintenance Organization (HMO) reported over 42 months, coupled with an online survey assessing CHP resuscitation knowledge, experience, training and self-confidence. Results: 22 resuscitations met inclusion criteria. In 21 CHP initiated chest-compressions but in only 8 cases they utilized the clinic’s automated external defibrillator (AED) prior to emergency medical services (EMS) arrival. There were 275 providers surveyed. Of the surveyed providers, 89.4% reported previous basic life support (BLS)/advanced cardiovascular life support (ALS) training, 67.9% within the last three years. Previous resuscitation experience was reported by 72.7%. The lowest scoring knowledge question was on indications for AED application −56.3%. Additionally, 44.4% reported low confidence in their resuscitation skills. CHP with previous cardiopulmonary resuscitation (CPR) experience reported higher confidence. Longer time since last CPR training lowered self-confidence. Conclusions: Early AED application is crucial for patients with OHCA. All clinics in our study were equipped with AED’s and most CHP received training in their use, but remained insecure regarding their use, often failing to do so.

## 1. Introduction

Out-of-hospital cardiac arrest (OHCA) is a major public health challenge [1,2,3]. Globally, the annual incidence of out-of-hospital cardiac arrest ranges from 20 to 140 per 100,000 people, with low survival rates from 2.0% to 16.3% [4,5,6,7,8]. In the United States, >500,000 persons every year experience a cardiac arrest, and survival rates are generally low <15.0% [9,10]. While most cases of out of hospital cardiac and respiratory arrests receive the initial professional care by emergency medical services (EMS) responders, some are initially cared for in primary health care.

Data from a US registry found that 2.0% of OHCA occurred in physician clinics [10]. The patients had a 26.9% survival rate, significantly higher than the 9.6% survival rate for the whole registry [10,11]. Additional published studies reported that 1.3% to 13.8% of the OHCA were initially cared for in community medical facilities [12,13,14,15]. Resuscitation in community clinics may occur in multiple scenarios and is affected by factors such as the clinic’s accessibility (including limited operating hours) and the availability of EMS. Patients may come to the primary physician/urgent care center and/or collapse in the clinic itself, because they are unaware of the severity of their symptoms. Additionally, they may experience life-threatening complications of care administered at the clinics. In rural communities, bringing the patient to the nearby clinic may be faster than awaiting EMS arrival.

Survival from cardio-pulmonary arrest is highly dependent on rapid provision of high quality BLS (basic life support) and early defibrillation, when indicated [10,16]. Therefore, relevant regulatory and professional guidelines for community healthcare facilities usually focus on promoting these two capabilities. As a result, community clinics in many countries are required to be prepared for resuscitations, by having the appropriate equipment and ensuring that their medical staff has the necessary knowledge and skills to perform high quality resuscitation.

Training and maintaining the knowledge and resuscitation skills of community healthcare facility staff is challenging, partly due to the low occurrence of these events and the ensuing lack of practical experience. Most studies have found that resuscitation training for nurses and physicians is plagued by low retention of knowledge and resuscitation skills, as well as by low levels of confidence in their ability to perform CPR [17,18].

Providing high quality CPR in community medical facilities is important in view of the small but significant proportion of OHCA occurring or being brought to them. For those patients the rapid availability of resuscitation equipment and trained community healthcare providers can be crucial for neurologically intact survival. The challenge of ensuring high quality CPR capabilities in all community clinics is significant. This is a classical example of low odds high stakes.

Primary care clinics, community specialty care centers and urgent care centers are operated in Israel by all major HMOs (health maintenance organizations) as well as private players. Maccabi Healthcare Services (MHS) is the second largest HMO in Israel, providing care to 2.3 million patients at 130 centers which include: 104 primary care clinics, 12 community urgent care centers and 14 specialty care centers. Clinics and centers are equipped with an AED and additional resuscitation equipment, in accordance with the Ministry of Health guidelines. The guidelines also require all physicians and nurses working in such clinics to undergo periodic CPR training (once every two years [19].

The current study reviewed actual resuscitations performed by community healthcare providers and reported to the MHS QA Department. Additionally, we evaluated multiple aspects of community healthcare provider preparedness for CPR, including CPR training, experience and confidence.

## 2. Materials and Methods

### 2.1. Study Design

The study included two separate components:

#### 2.1.1. Analysis of Reported Resuscitations

Data was obtained from compulsory electronic pre-formatted reporting forms of all resuscitation to Maccabi HMO Quality Assurance (QA) department. Following Institutional Review Boards (IRB) approval, we, thus, identified all reported resuscitations over a 42 month period, starting January 2014. Predefined data was collected from report sheets and MHS electronic medical records (EMRs), assessing case characteristics and the resuscitative actions by Maccabi HMO community nurses and physicians.

Patients who underwent resuscitation at primary care facilities were identified through the mandatory reporting forms sent to the HMO’s Quality Assurance (QA) department. In accordance with Maccabi’s procedures, each resuscitation event is reported using a dedicated digital form. This form is automatically sent by email to the Quality Assurance (QA) department for case-by-case validation. Additionally, event is reported in the patient’s electronic medical record (EMR).

Data was collected form resuscitation forms and patient’s EMR, using a structured from based on the Utstein guidelines [20]. Data fields included patient’s sex, age, patient’s presenting symptoms, relevant past medical history, current medications, location of arrest, resuscitative actions by clinic personnel (chest compressions, connection AED), return of spontaneous circulation (ROSC) and survival to hospital admission and discharge. Only patients with verified cardiac or respiratory arrest were included in the study. Patients for whom data was missing, were excluded from the study.

#### 2.1.2. The Questionnaire

An anonymous online questionnaire was created, validated and administered to physicians and nurses working in MHS community clinics and centers (Appendix A). An email with a link to the survey and a letter explaining the study goal was sent to all community healthcare providers who met the inclusion criteria. Potential participants were assured anonymity and confidentiality. All surveys were self-administered. The potential participants received two posts using each method, with a time lag approximating two weeks between the first and second post. The questionnaire was created using a modified Delphi process that included experts in primary care and resuscitation, thus, promoting both content and expert validity of the study tool. The questionnaire was generated through collaboration between an expert in family medicine (working in a representative primary care clinic), a representative of the EMS and an expert in intensive care, resuscitation and research, thus promoting both content and expert validity of the study tool. Following multi-disciplinary discussions, a questionnaire comprising 27-item was created for the purpose of this study. The clarity of the questions and their relevance was validated by two external consultants [21].

The 27-item questionnaire contained 3 demographic questions, 4 questions addressing professional education, 6 about CPR training and experience, 5 assessing CPR knowledge and 9 addressing provider’s level of confidence in their ability to perform high-quality resuscitations.

In the knowledge section, respondents were asked to answer five multiple choice questions regarding resuscitation, with each question having a single correct answer. The knowledge score of was calculated according to the number of correct answers.

A 6-point Likert scale was used to assess levels of confidence, ranging from totally disagree to totally agree (0 = totally disagree, 1 = slightly disagree, 2 = somewhat agree, 3 = moderately agree, 4 = strongly agree and 5 = totally agree)

An email explaining the study goal was sent to all community healthcare providers who met the inclusion criteria. Potential participants were assured anonymity and confidentiality. All surveys were self-administered.

The MHS institutional review board approved the study protocol prior to the initiation of data collection.

### 2.2. Statistical Analysis

Data entered and analyzed using SPSS (IBM SPSS Statistics for Windows, Version 21.0. IBM Corp, Armonk, NY, USA).

In the first study component, descriptive statistics were used to describe the patients’ and resuscitation characteristics.

In the second part, questionnaire: Missing responses to specific questions in the completed questionnaires were coded as missing. Notably, 2% (*n* = 5) of the questionnaires had 3 or more missing fields and were excluded.

Analysis of the questionnaires included descriptive statistics (e.g., number and percent of respondents who selected each response option). Percentages were calculated from the total number of respondents, including those with missing responses.

One-way ANOVA was used to test knowledge differences, level of confidence and influence of work settings. The independent samples t-test was used to test correlations between knowledge and level of confidence. Responders were categorized to two groups based on level of confidence in the ability to perform high quality CPR (based on the average score on these 9 questions). Low level of confidence was attributed to the score up to 2.9 and high level of confidence to those with higher average scores. Finally, a hierarchical linear regression model was used to predict level of confidence.

A *p*-value < 0.05 was considered significant.

## 3. Results

During the study period from January 2014 to July 2017 (42 months) 28 resuscitations were reported to the MHS Quality Assurance (QA) department, two were excluded due to missing or unclear data. Four resuscitations were reported by home care providers (out of 130,874 home care visits during the study period), nine by urgent care centers (out of 1,107,325 visits) and 15 by primary (nursing) care clinics (out of 4,392,429 patient visits). In order to estimate the incentive for every physician and nurses working in MHS community clinics to be involved in a resuscitation event we examined the number of visits during the study period.

Four of the resuscitations involved children (18 y or under), and 22 involved adults with an average age of 65.1 y. 14 of the 22 adult cases involved males.

As shown in Table 1, chest compressions were initiated by community healthcare providers in 21 of the 22 cases (21, 95.5%). In 8 of the 22 cases an AED was applied by the community healthcare providers. In 16 of the 26 patients (16, 61.5%) the time interval from the moment of identification that the resuscitation event occurs to the time the first emergency response was immediately. Ventricular fibrillation was reported in 6 of the 22 adult cases (6, 27.3%). %). In 16 of the 26 patients (16, 61.5%) medications were given during resuscitation, 11 of them treated by adrenaline. In 18 of the 26 patients ROSC was achieved, arriving alive to the hospital and being admitted (did not die in the emergency department).

Medical staff training, experience, knowledge and confidence in performing resuscitation skills.

In the second part of the study a total of 149 nurses and 126 doctors responded to an intranet administered questionnaire (out of 1280 nurses and 1490 doctors, response rate 10%). The responder’s demographic, education and professional characteristics were similar to those of the target group as a whole (Maccabi HMO community health care providers) (Table 2).

### 3.1. BLS/ACLS Training

Most respondents had undergone BLS training and/or ACLS training (253, 89.4%), most within the last one to three years (194, 67.9%). With reference to the question: “Would you be interested in additional resuscitation training and if so what type of training?”. More than half of the respondents replied that they would like to undergo a resuscitation workshop at the medical center where they work in conjunction with fellow co-workers and the clinic’s resuscitation equipment (167, 56.9%). Further, 125 respondents (42.6%) reported that they were interested in receiving CPR training more frequently than once every three years, while only 39 (13.3%) said that they did not feel the need for additional training.

### 3.2. Previous CPR Experience

Almost half (49.0%) the respondents had previous experience in resuscitation more than three years prior to the survey, and only 24.0% had no CPR experience (Figure 1).

Knowledge of the presence and location of resuscitation equipment in the clinics.

Respondents were asked about the CPR equipment in the medical center where they work and whether they have been trained in the use of this equipment. Most respondents stated that their clinic is equipped with resuscitation cart (83.6%) and 81.9% of the respondents knew of the existence of a defibrillator in their clinic. Moreover, 75.4% knew of the presence of a first aid bag. Respondents were also asked if they had attended any training on AEDs. Almost all respondents had undergone AEDs training (89.4%), most within the last three years (67.9%).

### 3.3. Resuscitation Guideline Knowledge

In the knowledge section, respondents were asked to answer five multiple choice questions regarding resuscitation with each question having a single correct answer. The knowledge score was calculated according to the number of correct answers. Only 37.0% (*N* = 103) of the respondents correctly answered all five questions, and 12.0% scored less than 60 (answered two or less correct answers). Table 3 shows most respondents knew the correct answer to most knowledge questions (56.3%–92.4%). The lowest scoring question was on the indications to connect an AED. Only 56.3% (*N* = 162) of the respondents answered correctly.

### 3.4. Confidence in Ability to Perform BLS and to Work as a Member of a Resuscitation Team

Regarding most statements in the questionnaire, the majority of participants reported a medium to high level of confidence, almost half 44.4% (*N* = 127) reported a low level of confidence. Further, 35.9% (*N* = 103) reported that they might hesitate to perform defibrillation, due to fear of harming the patient, and 36.9% (*N* = 106) reported that they might hesitate before commencing CPR.

### 3.5. Confidence in Performing Resuscitation Skills

A statistically significant correlation was found between work experience (years in the profession) and level of confidence (*p* < 0.01, *r* = 0.17).

The level of confidence significantly differed with gender (t (271) = 2.55, *p* = 0.01). Males scored significantly higher than women.

### 3.6. Work Settings

We found that those who working in high risk settings (urgent care center and community specialty centers) felt more confident in their resuscitation skills (F (3, 284) = 6.55, *p* = 0.000) and knowledge (F (3, 273) = 3.95, *p* = 0.009) than those working in lower risk settings (home care providers, primary (nursing) care clinics).

### 3.7. Previous Resuscitation Training

Respondents who reported having had CPR training last year reported higher level of confidence in the relevant statements compared to those whose CPR training had taken place more than three years ago (F (2, 285) = 6.08, *p* = 0.003).

### 3.8. Resuscitation Experience

Our study demonstrated that previous provider experience in care for cardiac arrests was significantly related to confidence in resuscitation skills (F (3, 284) = 20.86, *p* = 0.000) and knowledge (F (3, 275) = 6.08, *p* = 0.003). The highest levels of confidence were found in those who encountered cardiac arrest in the past year.

### 3.9. Prediction of Level of Confidence

To predict the level of confidence in performing resuscitation skills we used a hierarchical linear regression model. (Table 4). The model’s prediction was significant (F (10, 221) = 10.71, *p* = 0.00), with the predictive variables adding 33% to the explained variance of efficacy. In the first level, when demographic and work settings predictors were used to depict the relationship between variables: the work settings, additional work setting in a hospital, experience in CPR and previous resuscitation training had significant effects on the model. Work in low acuity settings {home care providers, primary (nursing) care clinics} was associated with a lower level of confidence compared to work in high acuity settings (urgent care center and community specialty centers) and additional work setting in a hospital. Those who had encountered more cardiac arrests, reported higher self-perceived confidence. Finally, prolonged period since the last CPR training lowered the level of confidence. In the second level, resuscitation knowledge as a predictor made no significant contribution to the model.

## 4. Discussion

This study assessed resuscitation in community healthcare facilities of a large HMO, by evaluating all reported resuscitations performed during a 2.5-year span and in parallel, assessing CPR training, knowledge, experience and level of confidence of the healthcare personnel working in these facilities. The incidence of OHCA in community health care facilities has only been sporadically studied.

Recently published studies from Singapore, Korea and other countries, reported that the incidence of OHCA in community healthcare facilities ranges wildly between 0.4% to 8.5% [22,23,24]. In a study of the Jerusalem district, carried out in 2004–2010, just 1.3% of cardiac arrests occurred in a primary care clinics [3]. This relatively low incidence may be explained by Jerusalem’s being is a major metropolitan area with several large and accessible hospitals. We assume that in rural communities, there will be a higher rate of patients seeking help from their primary physician/urgent care center for life threatening conditions and/or cardio-respiratory arrests in the clinic itself. Occurrence of cardiac arrest in a healthcare clinic setting is uncommon but not rare. A possible explanation for the very rare incidence in our community healthcare outpatient cohort could be that the majority of out-of-hospital cardiac arrests (OHCA) occurred at the patient’s home, additionally the most severe cases usually go directly to a hospital and some patients may come to the clinic primary physician/urgent care center and/or collapse in the clinic itself, because they are unaware of the severity of their symptoms. Therefore, resuscitation in community medical facilities utilizing the availability of healthcare professionals and resuscitation equipment may be an excellent opportunity to prevent death and disability. The community healthcare provider’s response to a cardiac arrest depends on multiple factors, including training and the availability of resuscitation equipment (most importantly an AED). In order to effectively respond to a cardiac arrest, both components are necessary.

Due to the retrospective nature of the study under-reporting was a concern. We attempted to calculate the expected number of OHCA cared for in community clinics. Based on a population of 2.2 million patients cared for by Maccabi HMO during 2014–2017, and an estimated incidence of 84/100,000 population per year we can expect 1848 cases of OHCA a year [25]. Assuming a similar rate of OHCA (1.3%) occurring in community health care facilities to that found in the Jerusalem study [12], we would expect 24 resuscitations to occur at Maccabi healthcare facilities per year. Our study though, found only an average of 8.5 such resuscitations per year. Under-reporting may partly explain this significant gap. Most reported resuscitations in our study occurred in primary (nursing) clinics (53.8%), all of which are equipped with resuscitation equipment, including an AED. Still, in only a minority of the 22 cases in which AED use was indicated it was applied by Maccabi clinic personnel (8, 36.4%), though chest compression CPR was provided to all cardiac arrest victims. Earlier AED applications may have further improved outcome by allowing defibrillation (when indicated) at an earlier stage and prior to EMS arrival [26]. The findings in our healthcare provider survey showed that most respondents had undergone AEDs training (89.4%), most within the last three years (67.9%). While most respondents knew the correct answer to most of the knowledge questions (56.3–92.4%), the lowest scoring question was on the indications to apply an AED. Only 56.3% (*N* = 162) of the respondents answered this question correctly. Moreover, in the questionnaire section dealing with provider level confidence and experience about half of the providers, 44.4% (*N* = 127) reported a low or very low level of confidence in their ability to adequately perform a resuscitation. Furthermore, 35.9% (*N* = 103) reported that they might hesitate to perform defibrillation because of fear of harming the patient and 36.9% (*N* = 106) reported that they may hesitate before commencing chest compressions. These findings may help explain the low AED application rates by community healthcare providers found in the first arm of our study and in other studies. A study from Singapore showed that 92.0% of general practitioners correctly identified defibrillation as one of the most important interventions in cardiac arrest, but only 60.0% reported that they knew how to operate an AED, and just 38.0% had attended training on AEDs. The cost and lack of confidence in using AEDs were the most commonly cited reasons why general practitioners did not have defibrillators in their clinics. In the study, only 27.0% reported that they had obtained an AED for their clinic, citing “lack of confidence” in their ability to operate it as a major barrier [7]. The current study highlights that despite the fact that CPR training was provided, and an AED was available, most community health care providers did not use it. The researchers believe that community health care providers’ response to OHCA needs to be improved, possibly through short but more frequent hands-on training, closely simulating their actual work environment and specifically aimed at improving AED utilization [27].

Limitations: This study has several limitations. Regarding the first part of the study in which predefined data was collected from report sheets and Maccabi’s EMR: This being a retrospective study, data quality may be affected by the quality of the community healthcare provider’s documentation of the patient’s medical record. Considering that some variables had missing data, it is possible that data were difficult to obtain in certain subgroups according to a Utstein data guidelines, such as 30-d survival or survival to discharge, neurological outcome at hospital discharge. Additionally, the number of resuscitation cases was low (as expected), requiring future studies to collaborate the findings. In the second part of the study, only nurses and physicians working in Maccabi HMO formal clinics were included. Many physicians in Maccabi HMO work in independent community clinics and were not included in the study. Finally, the questionnaire response rate was about 10% (lower than hoped) and although respondents’ characteristics were similar to the entire target population, some inaccuracies may have occurred. The study also evaluated a single large HMO, but its findings are supported by a limited number of studies performed in other countries with similar community health care systems.

## 5. Conclusions

Sudden cardiac arrest is uncommon in community settings but is a common cause for unexpected deaths. Early effective care (CPR and the use of an AED) can significantly decrease mortality. During the period January 2014 to July 2017 (42 months) 28 OHCA was reported to our register. Four of the resuscitations involved children (18 y or under), and 22 involved adults. In 21 cases (of the 22 adults (21, 95.5%)) CPR was started and in 8 cases (of the 22 adult (8, 36.4%)) an AED was attached. In the second part of the study a total of 149 nurses and 126 doctors responded to an intranet administered questionnaire (out of 1280 nurses and 1490 doctors, response rate 10%). Due to the very low number of answers the results are very insecure. The findings in our healthcare provider survey demonstrated a gap in both the knowledge (specifically regarding defibrillation, the single most effective intervention) and level of confidence in resuscitation skills. This gap may very well explain why despite the AED’s immediate availability, providers in our study, applied it in less than half the cases in which it was indicated.

### Implications for Practice

This study highlights the potential role of primary health care providers and clinics in OHCA. It also emphasizes the importance of CPR training and skills maintenance, to encourage early defibrillation by community healthcare providers and the failure of current methodology (similarly applied in almost all countries with modern community healthcare) to do so. Training may probably be the most significant parameter for a health care professional in order to be efficient during the management of cardiac arrest victims and to increase survival. The researchers believe that there is a need to develop short but more frequent hands-on training, closely simulating actual work community health care provider’s environment and specifically aimed at improving AED utilization.

## Figures and Tables

**Figure 1 ijerph-18-06612-f001:**
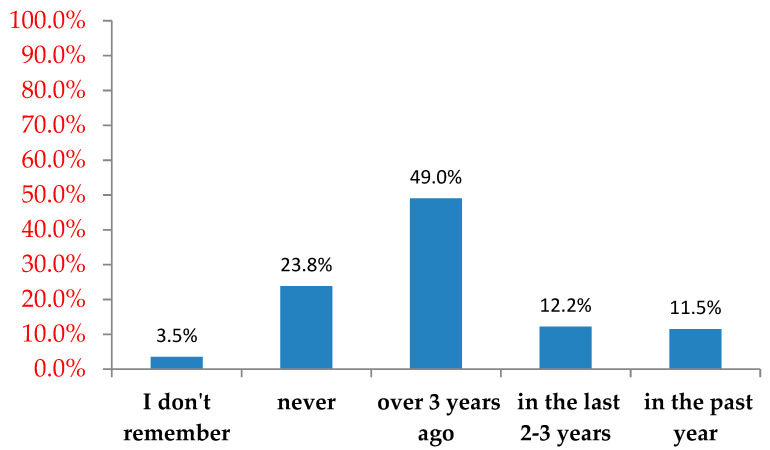
Previous CPR experience.

**Table 1 ijerph-18-06612-t001:** Characteristics of Cardiopulmonary resuscitation in adults.

Chest Compression Characteristics	Female Patients	Male Patients	Total
Physician led Chest compressions	6 (27.3%)	2 (25.0%)	4 (28.6%)
Nurse led chest compressions	9 (40.9%)	4 (40.0%)	5 (35.7%)
Leading chest compression provider not listed	1 (4.5%)	0	1 (7.1%)
Chest Compressions not performed by HMO staff	1 (4.5%)	1 (12.5%)	2 (14.3%)

**Table 2 ijerph-18-06612-t002:** Demographic characteristics of respondents (*n* = 275) and target group (*n* = 2800).

	Nurses		Physicians	
	Respondents	Target Group	Respondents	Target Group
	*N*-149	*N*-1300	*N*-126	*N*-1500
Age (years)	43.4	44.33	52	54.49
Mean (SD)	(9.87)	(8.59)	(10.6)	(12.3)
Experience (years)	15.6	11.03	21.2	12.56
Mean (SD)	(8.63)	(9.46)	(11.7)	(10.8)
Gender				
Female, *n* (%)	143 (95.9%)	(93.80%)	59 (48%)	579 (38.6%)
Male, *n* (%)	6 (4.1%)	(6.20%)	64 (52%)	921 (61.4%)

**Table 3 ijerph-18-06612-t003:** Distribution of knowledge question responses in all participants in absolute numbers and percentage scale.

Question	*N* (%)
Q1. A 62-year-old patient presented to your clinic and collapsed while waiting to be evaluated. You have connected the patient to the monitor—what is his cardiac rhythm (participants were presented with a rhythm strip of ventricular fibrillation and asked to identify the rhythm).	218 (75.7%)
Q2. Which of the following rhythm disturbances would you expect in an adult patient who suddenly collapses in front of you and is pulseless on evaluation?	253 (87.8%)
Q3. Which of the following is the major immediate risk in an adult unconscious patient?	202 (70.1%)
Q4. You are in your clinic (equipped with an Automated External Defibrilator). Which of the following actions would you perform on an adult patient who collapsed in front of you and is pulseless.	162 (56.3%)
Q5. Which of the following is correct regarding the defibrillation of a patient in ventricular fibrillation?	266 (92.4%)

**Table 4 ijerph-18-06612-t004:** Regression coefficients to predict the level of confidence in performing resuscitation.

Predictive Variables First Level	Coefficients					
	R^2^	*p*	t	B	SE	
Gender (1 = male)	0.33	0.14	1.46	0.16	0.11	0.10
Age		0.85	−0.18	−0.01	0.01	−0.02
Professional experience		0.28	1.06	0.01	0.01	0.12
Profession (1 = doctor)		0.14	−1.48	−0.15	0.10	−0.10
Work settings (1 = primary care clinics)		0.01	−2.53 *	−0.25	0.10	−0.14
Additional work setting in a hospital (1 = yes)		0.01	2.37 *	0.36	0.15	0.14
Experience in CPR		0.00	4.51 **	0.21	0.04	0.33
cardiac arrests in the past (1 = yes)		0.16	1.39	0.16	0.12	0.10
Previous resuscitation training		0.00	−3.14 **	−0.24	0.07	−0.18
**Second level**						
Level of knowledge	0.33	0.40	0.89	0.04	0.00	0.00

* *p* < 0.05, ** *p* < 0.01.

## Data Availability

All supporting data appears in the manuscript.

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
