# Peer review of "Resuscitation in Community Healthcare Facilities in Israel"

_ijerph, 2021, doi:10.3390/ijerph18126612_

Round 1

Reviewer 1 Report

Congratulations on an interesting study. The paper is well-written and interesting. However, I have several comments. Please do comment on them.

Point 1. All work. Please insert bibliographic references in square brackets, for example [2].

Point 2. Expand Abbreviations: BLS-D/ALS, AED, CPR, MHS QA, IRB, MHS' EMR, ED, ACLS.

Point 3. Please use the acronym BLS/AED and not BLS-D.

Point 4. Please forward appendix to the manuscript.

Point 5. Please indicate against which resuscitation guidelines (algorithms) the data obtained from the study were evaluated.

Point 6. Illegible Table 1. Characteristics of CPR in adults.

Point 7. Please standardize the nomenclature: Physician.

Point 8. Table 2. Demographic characteristics of respondents (N=275) and target group (N=2800). I don't understand this table and data. For what purpose do you provide the P-value?

Point 9. Figure 1. Previous CPR experience. It is better if the % scale is based on a range from 0 - 100%.

Point 10. The conclusions are very expansive. Please list some of the most important ones from your study.

Point 11. Please add and expand subsection 6. Implications for Practice

Point 12. References: Please review the literature and try to give full bibliographic data including doi access, for example:

Å»uratyÅ„ski, P.; ÅšlÄ™zak, D.; DÄ…browski, S.; Krzyżanowski, K.; MÄ™drzycka-DÄ…browska, W.; Rutkowski, P. Use of Public Automated External Defibrillators in Out-of-Hospital Cardiac Arrest in Poland. Medicina 2021, 57, 298. doi: 10.3390/medicina57030298

Point 13. References: to add more value to your work, please include a reference to the study: Å»uratyÅ„ski, P.; ÅšlÄ™zak, D.; DÄ…browski, S.; Krzyżanowski, K.; MÄ™drzycka-DÄ…browska, W.; Rutkowski, P. Use of Public Automated External Defibrillators in Out-of-Hospital Cardiac Arrest in Poland. Medicina 2021, 57, 298. doi: 10.3390/medicina57030298

Reviewer 2 Report

Good study. Please work on the conclusion additional recommendations concerning training should be made as it relates to improving OHCA.  

Reviewer 3 Report

Se my comments below as pdf

Round 2

Reviewer 1 Report

I thank the authors for addressing my suggestions and taking them into consideration. The authors have made changes that allow the publishing process to proceed. 
I recommend the revised manuscript for publication. 

Reviewer 3 Report

See previous discussion.

I am OK with the respons/Leif